# Blood M2-like Monocyte Polarization Is Associated with Calcific Plaque Phenotype in Stable Coronary Artery Disease: A Sub-Study of SMARTool Clinical Trial

**DOI:** 10.3390/biomedicines10030565

**Published:** 2022-02-28

**Authors:** Silverio Sbrana, Antonella Cecchettini, Luca Bastiani, Nicoletta Di Giorgi, Annamaria Mazzone, Elisa Ceccherini, Federico Vozzi, Chiara Caselli, Danilo Neglia, Alberto Clemente, Arthur J. H. A. Scholte, Oberdan Parodi, Gualtiero Pelosi, Silvia Rocchiccioli

**Affiliations:** 1CNR Institute of Clinical Physiology, 54100 Massa, Italy; luca.bastiani@ifc.cnr.it; 2Department of Clinical and Experimental Medicine, University of Pisa, 56126 Pisa, Italy; antonella.cecchettini@unipi.it; 3CNR Institute of Clinical Physiology, 56124 Pisa, Italy; digiorgi@ifc.cnr.it (N.D.G.); ceccherini@ifc.cnr.it (E.C.); vozzi@ifc.cnr.it (F.V.); chiara.caselli@ifc.cnr.it (C.C.); oberdan.parodi@virgilio.it (O.P.); pelosi@ifc.cnr.it (G.P.); 4Fondazione Toscana Gabriele Monasterio, 54100 Massa, Italy; mazzone@ftgm.it (A.M.); clemente@ftgm.it (A.C.); 5Fondazione Toscana Gabriele Monasterio, 56124 Pisa, Italy; dneglia@ftgm.it; 6Department of Cardiology, Leiden University Medical Center, 2333 ZA Leiden, The Netherlands; a.j.h.a.scholte@lumc.nl

**Keywords:** coronary artery disease, blood monocyte subsets, flow cytometry, plasma cytokines, coronary CT angiography, plaque calcium volume

## Abstract

Background: Atherosclerosis is a chronic inflammatory disease. The balance between pro- and anti-inflammatory factors, acting on the arterial wall, promotes less or more coronary plaque macro-calcification, respectively. We investigated the association between monocyte phenotypic polarization and CTCA-assessed plaque dense-calcium volume (DCV) in patients with stable coronary artery disease (CAD). Methods: In 55 patients, individual DCV component was assessed by quantitative CTCA and normalized to total plaque volume. Flow cytometry expression of CD14, CD16, CD18, CD11b, HLA-DR, CD163, CCR2, CCR5, CX3CR1 and CXCR4 was quantified. Adhesion molecules and cytokines were measured by ELISA. Results: DCV values were significantly associated, by multiple regression analysis, with the expression (RFI) of CCR5 (*p* = 0.04), CX3CR1 (*p* = 0.03), CCR2 (*p* = 0.02), CD163 (*p* = 0.005) on all monocytes, and with the phenotypic M2-like polarization ratio, RFI CCR5/CD11b (*p* = 0.01). A positive correlation with the increased expression of chemokines receptors CCR2, CCR5 and CX3CR1 on subsets Mon1 was also present. Among cytokines, the ratio between IL-10 and IL-6 was found to be strongly associated with DCV (*p* = 0.009). Conclusions: The association between DCV and M2-like phenotypic polarization of circulating monocytes indicates that plaque macro-calcification in stable CAD may be partly modulated by an anti-inflammatory monocyte functional state, as evidenced by cell membrane receptor patterns.

## 1. Introduction

It is well known that inflammation plays a key role in the pathogenesis of atherosclerotic coronary artery disease (CAD), which can be considered as a chronic non-resolving inflammatory disease of the vessel wall, characterized by the continuous interaction between systemic pro- and anti-inflammatory factors [1]. This balance is associated with a low-grade systemic immune-inflammation and can be assessed by measuring several blood markers, some of which have also been proposed as candidate predictors of functionally severe coronary stenosis in stable CAD [2]. 

In fact, both the degree of plaque-related coronary stenosis and plaque composition would depend on two opposite and simultaneous processes: newly formed atherosclerotic tissue formation—due to inflammation/endothelial activation—and anti-inflammatory/pro-reparative evolution of already formed plaques. The latter is at the origin of plaque macro-calcification, characterized by dense-calcium large deposits [3], detectable by Computed Tomography Coronary Angiography (CTCA) [4,5,6]. Circulating monocytes and arterial wall macrophages play a key role in these processes [7], closely linked not only to their overall functional polarization and phenotype, but also to the relative phenotypic and functional ratios among the different cellular subsets [8,9].

Circulating monocytes represent a heterogeneous population composed of at least three main subsets identified as Mon1 (CD14++/CD16- or “classical”), Mon2 (CD14++/CD16+ or “intermediate”) and Mon3 (CD14+/CD16++ or “non-classical”) [8].

The functional interaction between tissue inflammatory aspects, medullary microenvironment and circulating blood is mediated by signals leading to a “*cellular training*” [10,11] of the monocyte cell development line; the terms *“M1-like”* and *“M2-like”* are used to illustrate the opposing activities of killing (pro-inflammatory, classically activated, “killer M1”) and repairing (anti-inflammatory, alternatively activated, “builder M2”) within a single immunological continuum, characterizing both circulating monocytes and tissue macrophages [12].

We have previously demonstrated that stable CAD patients under optimal medical therapy undergo a net progression of CTCA-assessed calcified plaque volume over time, suggesting a statin-driven plaque stabilization [5]. In this study, we aimed to investigate, in the same subgroup of stable CAD patients that were enrolled in the SMARTool clinical trial of our previous report [13], whether the CTCA-assessed fractional volume of calcium was cross-associated with a phenotypic monocyte polarization typical of a systemic anti-inflammatory/pro-reparative state. The polarization state of circulating monocytes can be determined by a distinctive membrane receptor pattern [12,14,15], as we previously reported for plaque stenosis severity, and was significantly associated with the increased expression of single surface markers CX3CR1, CCR2, CCR5 and CD163 on subset Mon2 [13]. However, the absolute expression of single markers could not fully reflect the actual pro-/anti-inflammatory balance. Thus, in this study, we have used some monocyte receptor ratios from all CD14++/+ cells. In particular, the expression of CD11b, an integrin molecule known to be up-regulated in a prevalent pro-inflammatory context (M1-like phenotypic polarization) [16], has been related to that of receptors CX3CR1, CCR2, CCR5 and CD163, known to be up-regulated under prevalent anti-inflammatory conditions (M2-like phenotypic polarization) [17,18,19]. 

In particular, we statistically evaluated, on a per-patient basis, the association between phenotypic features of blood monocytes, assessed by flow cytometry quantification of CD14, CD16, CD18, CD11b, HLA-DR, CXCR4 and of CD163, CCR2, CCR5, CX3CR1 expressions—as well as their ratio to relative fluorescence intensity (RFI) of CD11b—and the global dense-calcium volume (DCV) of all coronary plaques in each patient, estimated by quantitative CTCA image analysis and normalized to total plaque volume.

## 2. Materials and Methods

### 2.1. Patients

Patients (*n*° = 73) were initially recruited at the Cardiology and Cardiovascular Medicine Division of Fondazione Toscana “G. Monasterio” (Pisa, Italy) from September 2016 to November 2017, within the clinical trial of 2020 EU Project SMARTool (ClinicalTrials.gov Identifier: NCT04448691). In brief, following the approval of the Ethical Committee of the “Area Vasta Nord-Ovest” of Tuscany Region (Italy), and patient written informed consent, a Caucasian population of male (*n*° = 48) and female (*n*° = 25) subjects, aged 48–82 years was included in the SMARTool observational cross-sectional sub-study to receive a CTCA scan [13]. In the present study, the evaluation had been restricted to those patients with detectable plaques that were available for quantitative analysis of dense-calcium volume (*n*° = 61). All these patients were on statin treatment and were submitted to a clinical follow-up to verify that their clinical conditions were stable in the last six months. Cumulative patients’ clinical characteristics, inclusion and exclusion criteria, as well as CTCA inclusion criteria and scan protocol, have been reported (see Appendix A) [13]. The administration of costicosteroid therapy at the time of blood collection was considered an additional exclusion criterion in addition to those previously indicated [13].

### 2.2. CTCA and Quantitative Image Analysis

CTCA was performed according to a predefined standard operating procedure to ensure optimal image quality [5]. All images were analyzed blinded to clinical data by a separate core laboratory (Leiden University Medical Center). Coronary arteries were assessed according to the modified 17-segment American Heart Association classification. First, a visual analysis was performed to assess the presence, location, severity and composition of coronary plaques. Subsequently, quantitative CTCA analysis was performed for all visually determined plaques, using a dedicated software package (Medis, QAngio CT Research Edition version 3.1.2.0). The complete workflow of quantitative CTCA analysis has previously been described in detail [6]. In brief, the 3-dimensional coronary tree was extracted from the coronary CTCA data set and straightened multi-planar reconstructions were created of each coronary artery. Subsequently, the lumen and vessel wall contours were automatically detected and these were manually adjusted if needed. Each atherosclerotic lesion was detected based on the lumen and vessel wall contours and the corresponding references lines, which indicate the normal tapering of the coronary artery. For each coronary lesion, stenosis parameters were calculated at the level of the minimal lumen area. In addition, total plaque volume and plaque volume according to the plaque composition were determined using predefined intensity cut-off values in Hounsfield units (HU): −30 to 75 HU for necrotic core plaque, 75 to 130 HU for fibro-fatty plaque, 130 to 350 HU for fibrous plaque and >350 HU for dense-calcium plaque [20]. 

CTCA-assessed dense-calcium fractional volume (DCV, a.u.) was estimated on a per-patient basis as the ratio of the sum of dense-calcium volume values, expressed in mm^3^ in all detected plaques (HU > 350), to the sum of total plaque volume values (mm^3^). 

### 2.3. Biochemical Analyses

In addition to the plasma analytes determined and described in our previous study [13] (glucose (mg/dL), creatinine (mg/dL), acid uric (mg/dL), total cholesterol (mg/dL), HDL-cholesterol (mg/dL), LDL-cholesterol (mg/dL), triglycerides (mg/dL), fibrinogen (mg/dL), Hs-CRP (mg/dL), ICAM-1 (ng/mL), VCAM-1 (ng/mL), IL-6 (pg/mL), IFN-γ (pg/mL), TNF-α (pg/mL) and IL-10 (pg/mL)), the plasma levels of MCP-1 (pg/mL), IL-8 (pg/mL), RANTES (pg/mL) and Fractalkine (pg/mL) were also determined by enzyme-linked immunosorbent assays (ELISAs), according to the manufacturer’s instructions (Thermo Fisher Scientific, MA, USA). The following commercial ELISA kits were used: BMS281 for human MCP-1 analysis; KHC0081 for human IL-8; EHRNTS for RANTES and EHCX3CL1 for human Fractalkine.

### 2.4. Flow Cytometry Analysis

Within 1 h after EDTA-anticoagulated blood collection, flow cytometry monocyte expressions of CD14, CD16, CD18, CD11b, HLA-DR, CD163, CCR2, CCR5, CX3CR1 and CXCR4 was quantified, both as percentage of positivity (%+) and relative fluorescence intensity (RFI), by using a lyse-no-wash three-color staining procedure, as previously described [13]. The modulation of the expression of the aforementioned monocyte markers, although initially selected for an evaluation of their role in the severity of coronary stenosis, can also provide information regarding the effect of the systemic inflammatory balance on the relative composition of coronary atherosclerotic plaques.

### 2.5. Statistical Analysis

Continuous data were presented as mean ± mean standard error (SEM). The comparison between groups was carried out by ANOVA (with Bonferroni’s correction) for continuous data (see Table 1). Multiple linear regression analysis was performed assuming DCV as the dependent variable. The independent variables used for the adjustment were continuous (either as source data or after appropriate numerical transformation of a nominal variable), and have been chosen for their known pathophysiological relevance in the initiation and progression of atherosclerotic disease. The model 1 adjustment (see Table 2) included Framingham Risk Score, diabetes, creatinine, endothelial activation (plasma levels of ICAM-1 and VCAM-1), systemic pro-/anti-inflammatory environment (plasma levels of Hs-CRP, IL-6, IFN-γ, TNF-α, IL-8, MCP-1, RANTES, Fractalkine, IL-10), dosage of statin therapy (mg/die), and use of oral antidiabetics. We would like to point out that the evaluation of the sex-related differences of the parameters included in the above multiple regression analysis model are automatically considered on the basis of the calculation of the Framingham Risk Score (a.u.). The relationships between DCV values and monocyte phenotypic features and IL-10/IL-6 ratio have been evaluated by multiple regression analysis (model 1 adjusted: see Table 2, Table 3, Table 4 and Table 5). Bivariate correlation analyses have been also used to investigate either the relationships between the monocyte phenotypic features that at multiple regression correlated significantly with DCV (see Appendix A), or the correlations between monocyte markers expression and circulating cytokines levels. All statistical analyses have been performed by Stat View 5.0 software program (SAS Institute, Cary, NC, USA). A *p* value < 0.05 was considered statistically significant.

## 3. Results

### 3.1. Patients Clinical Characteristics and Plasma Biochemistry

Demographic, clinical and laboratory characteristics of all enrolled patients have been reported in our previous paper [13]. The most relevant clinical and biohumoral parameters used in this study for adjustment for multiple regression analysis in those patients with detectable coronary plaques (*n*° = 61) are reported in Table 1 according to CAD severity [21,22], classified as (i) CAD1 (maximal stenosis < 25%), (ii) CAD2 (25% ≥ maximal stenosis < 50%) and (iii) CAD 3 (maximal stenosis ≥ 50%). 

Mean DCV values are also reported in the last row of the table to show their association with CAD stenosis severity: significantly higher values are found in CAD3 vs. CAD1 (ANOVA *P =* 0.02).

**Table 1 biomedicines-10-00565-t001:** Clinical and biohumoral parameters in all patients and by CAD severity classes.

	All Patients (*n*° = 61)	CAD1 (*n*° = 19)	CAD2 (*n*° = 21)	CAD3 (*n*° = 21)	ANOVA *P*
Age (years)	68.7 ± 1.0	66.37 ± 2.06	70.24 ± 1.72	69.19 ± 1.41	Ns
Gender (M/F, *n*°)	44/17	14/5	12/9	18/3	Ns
Framingham Risk Score (a.u) (FRS)	15.36 ± 0.43	14.44 ± 1.09	16.09 ± 0.56	15.33 ± 0.65	Ns
Diabetes, *n*° (%)	20 (32.79)	3 (4.92)	6 (9.84)	11 (18.03)	0.0419 *
Oral antidiabetics, *n*° (%)	18 (29.51)	3 (4.92)	5 (8.20)	10 (16.40)	Ns
Statin therapy (dosage, mg/die)	13.03 ± 1.37	9.47 ± 1.95	13.09 ± 2.73	16.19 ± 2.20	Ns
Creatinine (mg/dL)	0.85 ± 0.03	0.87 ± 0.04	0.78 ± 0.04	0.90 ± 0.04	Ns
ICAM-1 (ng/mL)	224.85 ± 12.94	247.36 ± 22.33	222.04 ± 21.22	208.38 ± 23.70	Ns
VCAM-1 (ng/mL)	641.10 ± 21.21	724.50 ± 53.35	547.93 ± 17.11	662.79 ± 25.85	0.0018 ^§^^
Hs-CRP (mg/dL)	0.44 ± 0.09	0.55 ± 0.17	0.33 ± 0.07	0.45 ± 0.21	Ns
IL-6 (pg/mL)	1.01 ± 0.12	1.26 ± 0.25	0.66 ± 0.10	1.13 ± 0.22	Ns
IL-10 (pg/mL)	27.21 ± 1.67	40.02 ± 2.91	23.70 ± 2.51	20.95 ± 1.44	<0.0001 ^§^*
IFN-γ (pg/mL)	32.29 ± 1.66	34.11 ± 4.52	30.52 ± 1.97	32.67 ± 2.44	Ns
TNF-α (pg/mL)	69.89 ± 2.96	73.33 ± 8.71	67.46 ± 4.03	69.71 ± 2.97	Ns
IL-8 (pg/mL)	2.02 ± 0.24	2.10 ± 0.47	1.56 ± 0.35	2.48 ± 0.41	Ns
MCP-1 (pg/mL)	176.24 ± 8.74	191.91 ± 12.28	177.59 ± 13.98	158.14 ± 18.82	Ns
RANTES (pg/mL)	146.65 ± 14.28	157.76 ± 24.90	144.77 ± 26.87	137.11 ± 22.48	Ns
Fractalkine (pg/mL)	0.96 ± 0.20	1.07 ± 0.32	1.32 ± 0.40	0.42 ± 0.24	Ns
**DCV (a.u.)**	**0.15 ± 0.01**	**0.12 ± 0.02**	**0.15 ± 0.02**	**0.19 ± 0.02**	**0.0204 ***

Data are presented as mean ± SEM (standard error of the mean) or as number (*n*°) and percentage (%), when appropriate. The Bonferroni post hoc (ANOVA *P*): * CAD1/CAD3, ^§^ CAD1/CAD2 and ^ CAD2/CAD3; *p* < 0.05: statistically significant; Ns: not significant; a.u. = arbitrary units.

The multiple regression analysis results between the same parameters—excluding gender and age, included in the computation of the Framingham Risk Score—and DCV values are reported in Table 2. The statistical method of DCV-associated multiple regression analysis requires a complete matching between all the biohumoral and clinical parameters of Table 2, thus, reducing the initial number of 61 to 55 patients analyzed. 

**Table 2 biomedicines-10-00565-t002:** Multiple regression between clinical and biohumoral parameters and DCV (model 1 adjustment).

	DCV (a.u.) (*n*° = 55)	
	Regression Coefficient	*p*-Value
Framingham Risk Score (a.u.)	−0.001	0.7427
Diabetes	0.099	0.1694
Oral antidiabetics	−0.109	0.1539
Statin therapy (mg/die)	−3.169 × 10^−4^	0.7898
Creatinine (mg/dL)	−0.015	0.8359
ICAM-1 (ng/mL)	9.678 × 10^−5^	0.4985
VCAM-1 (ng/mL)	5.841 × 10^−5^	0.5352
Hs-CRP (mg/dL)	−0.047	0.0216 *
IL-6 (pg/mL)	0.042	0.0281 *
IL-10 (pg/mL)	−0.002	0.0633
IFN-γ (pg/mL)	−0.002	0.1115
TNF-α (pg/mL)	4.030 × 10^−4^	0.4824
IL-8 (pg/mL)	−0.001	0.8409
MCP-1 (pg/mL)	−3.021 × 10^−4^	0.1961
RANTES (pg/mL)	−2.785 × 10^−5^	0.8319
Fractalkine (pg/mL)	0.001	0.9250

** p* < 0.05: statistically significant (by multiple regression analysis, model 1 adjustment); a.u. = arbitrary units.

A negative statistically significant association between Hs-CRP and DCV and a positive association with IL-6 were observed. In addition, the ratio IL-10/IL-6 (51.63 ± 5.71, mean ± SEM)—not shown in the table—was found to be the most significantly associated, among all biohumoral and clinical parameters, with DCV (*P* = 0.009).

### 3.2. Relationship between Monocyte Cell Count, Phenotypic Features and DCV

Neither total cell count of all CD14++/+ monocytes, nor their fractions (%) and absolute numbers (*n*° of cells/μL) of circulating monocyte subsets were significantly associated at multiple regression analysis with DCV values. The expression levels of blood CD14++/+ monocyte surface markers that were positively and significantly associated by multiple regression analysis with DCV values, are reported in Table 3.

**Table 3 biomedicines-10-00565-t003:** Significantly positive association, at multiple regression analysis, between blood CD14++/+ monocyte surface markers and DCV.

All CD14++/+ Monocytes		DCV (a.u.) (*n*° = 55)
**CCR5**	%+	*p* = 0.0115 *
	RFI	*p* = 0.0452 *
CX3CR1	RFI	*p* = 0.0309 *
CCR2	RFI	*p* = 0.0226 *
CD163	RFI	*p* = 0.0054 *

** p* < 0.05: statistically significant (by multiple regression analysis, model 1 adjustment); %+ = percentage of positivity; RFI = relative fluorescence intensity; a.u. = arbitrary units.

For the Mon1 subset, the surface markers showing a significantly positive association with DCV were CCR5 (%+, *p* = 0.01; RFI, *p* = 0.05), CX3CR1 (%+, *p* = 0.05; RFI, *p* = 0.03) and CCR2 (%+, *p* = 0.002; RFI, *p* = 0.01).

For the Mon2 subset, only CCR5 expression exhibited a positive association with DCV values, but only as %+ (*p* = 0.05).

For the Mon3 subset, no associations were found between markers expression and DCV values.

The evaluation by multiple regression analysis of the reciprocal receptor expression ratios between the subsets showed a significantly positive association of DCV with a higher level of functional activation, as well as of pro-adhesive and chemokine-driven pro-migratory capacity of subsets Mon1 and Mon2, when compared with those of subset Mon3 (Table 4).

**Table 4 biomedicines-10-00565-t004:** Multiple regression of the ratio of monocyte subsets’ markers expression and DCV.

		DCV (a.u.) (*n*° = 55)
**Ratio Mon1/Mon3**	Ratio of %+	CX3CR1 (*p* = 0.0293) * CCR2 (*p* = 0.0041) *
	Ratio of RFI	HLA-DR (*p* = 0.0475) * CD11b (*p* = 0.0387) * CCR2 (*p* = 0.0020) *
**Ratio Mon2/Mon3**	Ratio of %+	CX3CR1 (*p* = 0.0269) * CCR2 (*p* = 0.0034) *
	Ratio of RFI	CCR2 (*p* = 0.0291) * HLA-DR (*p* = 0.0235) *

* *p* < 0.05: statistically significant (by multiple regression analysis, model 1 adjustment); %+ = percentage of positivity; RFI = relative fluorescence intensity; a.u. = arbitrary units.

### 3.3. Associations of Monocyte Phenotypic Ratios with DCV

Up to now, the systemic pro-/anti-inflammatory balance characterizing clinical conditions such as diabetes, hypercholesterolemia and atherosclerosis have been studied by means of the numerical ratio between monocyte fractions with different functional receptorial polarization [23,24]. However, in our study, we have evaluated the relative expression (ratio) of monocyte markers with opposite functional regulation at the single cell level.

The associations, observed at multiple regression analysis, between the phenotypic polarization ratios (RFI) on all CD14++/+ populations and DCV are shown in Table 5.

**Table 5 biomedicines-10-00565-t005:** Multiple regression of RFI ratio of opposite polarized monocyte markers and DCV.

		DCV (a.u.) (*n*° = 55)	
		Regression Coefficient	Capacity-Value
Ratio CD11b/CD163	Ratio of RFI	--------------	
Ratio CD163/CD11b	Ratio of RFI	0.454	0.0074 *
Ratio CD11b/CX3CR1	Ratio of RFI	--------------	
Ratio CX3CR1/CD11b	Ratio of RFI	0.816	0.0111 *
Ratio CD11b/CCR5	Ratio of RFI	--------------	
Ratio CCR5/CD11b	Ratio of RFI	0.999	0.0136 *
Ratio CD11b/CCR2	Ratio of RFI	--------------	
Ratio CCR2/CD11b	Ratio of RFI	0.490	0.0232 *

** p* < 0.05: statistically significant (by multiple regression analysis, model 1 adjustment); RFI = relative fluorescence intensity; a.u. = arbitrary units.

Among the monocyte phenotypic ratios reported in Table 5, those with the highest significant positive regression coefficient have been chosen as identifiers of the systemic monocyte polarization associated with calcified plaque volume. In particular, the ratio, RFI CCR5/CD11b was chosen for the M2-like polarization (regression coefficient = 0.999), while the ratio, RFI CD11b/CD163 was chosen for the M1-like polarization (regression coefficient = 0.177). Moreover, while the phenotypic ratio, RFI CCR5/CD11b correlated positively with almost all the monocyte phenotypic features that, at multiple regression, resulted positively associated also with DCV (see Appendix A), the phenotypic ratio, RFI CD11b/CD163 correlated positively only with the expression level (RFI) of the CD14 molecule on the entire monocyte population (*p* < 0.0001; R = 0.444). 

## 4. Discussion

### 4.1. Study Results

In this study, DCV was found to be independently associated, in addition to the monocyte expression level of the M2 polarization marker, CD163, with an increased expression of surface receptors for the chemokines, MCP-1, RANTES and Fractalkine, on all CD14++/+ monocytes and on Mon1 subset. Additionally, the M2-like polarization ratio, RFI CCR5/CD11b, independently associated with DCV, was positively and significantly associated with the expression levels of all these surface receptors, suggesting an M2 polarization-mediated effect, promoting the survival and accumulation of monocytes migrated from blood into the vessel wall. In fact, as known from the literature [18], the aforementioned monocyte surface molecules show a divergent regulation of expression depending on the prevailing inflammatory environment M1-like (pro-inflammatory) or M2-like (anti-inflammatory). Molecules such as CCR2, CCR5 and CD163 are up-regulated in the presence of anti-inflammatory signals (such as IL-10) and down-regulated by pro-inflammatory stimuli (such as IFN-γ, TNF-α and IL-1β). 

### 4.2. Comparison with Similar Studies

Monocyte polarization has been assessed so far in terms of ratio between cellular fractions identified through the use of polarization-specific markers [12,17]. In the present study, we evaluated the monocyte polarization state at single cell level by simultaneously quantifying the relative expression of polarization markers with opposite functional regulation and calculating their ratio; this was a better index of the actual balance between pro- and anti-inflammatory stimuli acting within a particular immunological environment. 

Our data suggest that the M2-like systemic inflammatory polarization, mainly associated with a calcific plaque phenotype, determines, through the up-regulation of some of the main chemokine receptors such as CCR5, CCR2 and CX3CR1, an increase in the migratory capacity and accumulation over time of circulating monocytes within the vascular wall. In particular, the correlation observed between the M2-like ratio, RFI CCR5/CD11b and the expression level (RFI) of Fractalkine receptor CX3CR1 could suggest a M2 polarization-mediated effect in favor of a higher accumulation over time of viable monocyte cells within the vessel wall [25]. 

This could amplify the mechanism of efferocytosis inside the plaque [26] and favor, through a positive amplification loop of the M2 functional polarization of infiltrating macrophages, the resolution of inflammation and the formation of large calcium deposits. 

The cytokine ratio, IL-10/IL-6 has been previously reported as a marker of immunosuppression associated with malignant tumors and/or with intense physical exercise [27,28]. We observed a close association between this ratio and the prevalence of dense-calcium plaque phenotype, suggesting that it may also be a useful marker in more complex immunological settings.

Furthermore, based on the significant positive correlations observed in our study between blood monocyte subsets phenotypic ratios and DCV, we can hypothesize that the M2-polarization-induced monocyte trans-endothelial migration is primarily sustained by circulating subsets Mon1 and Mon2. 

While the differential contribution of vascular wall M1-like or M2-like polarized macrophages to VSMC osteogenic differentiation has been already reported [3], our data provide the first demonstration of a direct association between monocyte phenotypic polarization and calcified plaque volume assessed by quantitative CTCA analysis in stable CAD patients under statin treatment.

### 4.3. Limits of the Study

The statistical significance of the associations between circulating monocytes and plaque calcium may be affected by the low number of patients included. In this proof-of-concept study, we evaluated a restricted cohort of patients undergoing a CTCA scan and monocyte analysis on fresh blood at the same time. 

Furthermore, given the small number of patients studied, it was not possible to perform a sex-subgroup-based statistical evaluation.

Further studies on larger patient populations are needed to identify monocyte receptors as biomarkers of calcific coronary plaque phenotype in CAD patients. 

Besides large dense-calcium deposits, the dense fibrous tissue component of coronary plaque can result from an anti-inflammatory reparative process. However, we conservatively chose not to quantify the fibrous component, due to the difficulty of isolating it from the other non-calcific components and the wide threshold (130 to 350 HU) to be adopted for its quantitative analysis.

The use of monocyte phenotypic polarization ratio proposed in this study requires further validations in a larger number of patients: testing using other monocyte markers which are more specific of an M2- or an M1-like polarization is also necessary to demonstrate the strength of our methodological approach.

## 5. Conclusions

This proof-of-concept study evaluates the association between circulating monocyte fingerprint and coronary plaque calcification.

The main findings of the study are: (1) The anti-inflammatory (M2-like) monocyte phenotypic polarization, with an increased relative expression (RFI) of CD163 and of chemokine receptors, was significantly associated with higher DCV values. (2) The plasma cytokine ratio IL-10/IL-6 showed a significant positive association with dense-calcium volume values, further supporting the involvement of an M2-like systemic polarization in plaque calcification. (3) A highly significant positive association was found between CTCA-assessed DCV values and RFI CCR5/CD11b ratio.

These findings demonstrate that quantitatively assessed dense-calcium plaque volume is significantly associated with an M2-like circulating monocyte fingerprint, best expressed by the ratio, (RFI) CCR5/CD11b.

In conclusion, a systemic environment with an anti-inflammatory monocyte functional polarization could facilitate the deposition of dense macro-calcifications in coronary plaques of CAD patients under statin treatment, thus, promoting their stabilization.

## Data Availability

Not applicable.

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
