# Peer review of "Blood M2-like Monocyte Polarization Is Associated with Calcific Plaque Phenotype in Stable Coronary Artery Disease: A Sub-Study of SMARTool Clinical Trial"

_biomedicines, 2022, doi:10.3390/biomedicines10030565_

Round 1

Reviewer 1 Report

In this manuscript, Silverio Sbrana et al. describe a positive correlation with the increased biomarker express modulated by an anti-inflammatory monocyte functional state in the blood of the patients with DCV

Introduction:

The introduction shown is in accordance with the study.

Material and method: The design of the material and method should take into account the following data.

Description of type of clinical trials

Description of the criteria for selecting patients for the study.

Why have the following measured parameters been selected?

Results: The design of the results should take into account the following data:

Include a brief description of the meaning of "Mon1", "Mon2", etc.

Include a brief bibliographic description of the meaning and selection of the different biomarker ratios selected and their significance.

Show the results obtained according to the sex subgroup, as there is a higher prevalence of atherosclerosis disease in the female sex.

Include some selected in CTCA and quantitative image analysis section.

Discussion:

The point of view should be modified and completed because it should not include conclusions in this section; however, it should justify the selection of each parameter measured in the study and explain the result obtained.

Author Response

Materials and Methods section:

(a) Description of type of clinical trials.

Reply: SMARTool Clinical trial is an international interventional trial recruiting patients in the framework of an H2020 funded project (PN689068). The patients are stable CAD patients with criteria reported in Table S2

(b) Description of the criteria for selecting patients for the study.

Reply: An additional table (Table S2 in Supplementary Materials) has been added containing the complete list of the inclusion and exclusion criteria used for the trial. In this paper the SMARTool population is restricted to one clinical center enrollments (FTGM-Italy) due to the need of fresh blood to be processed.

(c) Why have the following measured parameters been selected?

Reply: the usefulness of the studied flow cytometric markers, both in relation to the systemic inflammatory polarization and to the plaque composition, was highlighted in the Flow Cytometry Analysis Section.

Results section:

(a) Include a brief description of the meaning of “Mon1”, “Mon2”, etc.

Reply: A brief description of monocyte subsets Mon1, Mon2 and Mon3 has been added to the introduction section.

(b) Include a brief bibliographic description of the meaning and selection of the different biomarker ratios selected and their significance.

Reply: A short bibliography describing the monocyte phenotypic ratios currently used in different clinical conditions, as well as the novelty and significance of the ratios introduced for the first time in our study, have been reported in the paragraph 2.3. (Results section) and in the paragraph 3.1. (Discussion section). 

(c) Show the results obtained according to the sex subgroup, as there is a higher prevalence of atherosclerosis disease in the female sex.

Reply: In paragraph 4.5. (Statistical Analysis) we added the following sentence: We would like to point out that the evaluation of the sex-related differences of the parameters included in the multiple regression analysis shown in Table 2 are already considered on the basis of the calculation of the Framingham Risk Score (a.u.).

Moreover, in the paragraph 3.3. (Limits of the study), we added a sentence regarding the inability to perform a sex-based assessment due to the low number of female patients recruited into the study.

Discussion section:

(a) The point of view should be modified and completed because it should not include conclusion in this section; however, it should justify the selection of each parameter measured in the study and explain the result obtained.

Reply: We modified section 3 (Discussion) by removing the paragraph concerning the main findings of the study and placing it within section 5 (Conclusions). Furthermore, in paragraph 3.1. (Study results) and in the paragraph 4.4 (Flow Cytometry Analysis) we highlighted the usefulness of the studied flow cytometric markers, both in relation to the systemic inflammatory monocyte polarization and to the coronary plaque composition.

Reviewer 2 Report

Blood M2-like monocyte polarization is associated with calcific plaque phenotype in stable coronary artery disease: a sub-study of SMARTool clinical trial 

I found this paper to be, overall, clearly written and in a coherent logic. All statistical or analytic procedures in this manuscript appear accurate to me. There are some minor places where the paper can be modified to read a little easier for people who are not clinical/statistical focused. 

  1. [page 4] this is where the notation "Mon1" ... "Mon3" first appeared, it is not clear to me what does it stand for, whether it refers to a previously discovered group or it is a group that was categorized by this paper. If you do a word search, this (page 4) is the first time the word "Mon1" appears and it can certainly use some explanation.
  2. It is not clear to me whether the authors distinguished circulating macrophages from monocytes; it appears that when authors used M2 categorization, it refers to alternatively polarized macrophages. 
  3. The M1 and M2 categorization can be further defined or clarified by using other sets of M1/M2 markers, but again this might be affected by the low patient number. 

Author Response

Point 1:

Reply: A brief phenotypic description of the three main human monocyte subsets Mon1, Mon2 and Mon3 has been added to the Introduction section.

Point 2:

Reply: In the Introduction we distinguished between circulating monocytes and tissue macrophages, although indicating the close relationship between the two cell types, in particular between circulating monocytes and macrophages of the vascular wall. We also indicated that the two cell types share similar functional polarizations M1-like (pro-inflammatory, classically activated) or M2-like (anti-inflammatory/pro-reparative, alternatively activated).

Point 3:

Reply: The need to test other more specific markers of monocyte polarization was indicated in section 3.3. (Limits of the study).

This manuscript is a resubmission of an earlier submission. The following is a list of the peer review reports and author responses from that submission.